# Revealing the molecular interplay of coverage, wettability, and capacitive response at the Pt(111)-water solution interface under bias
Federico Raffone[1], Rémi Khatib[2,4], Marialore Sulpizi[3] & Clotilde Cucinotta [1] ✉

While electrified interfaces are crucial for electrocatalysis and corrosion, their molecular morphology remains largely unknown. Through highly realistic ab initio molecular dynamics simulations of the Pt(111)-water solution interface in reducing conditions, we reveal a deep interconnection among electrode coverage, wettability, capacitive response, and catalytic activity. We identify computationally the experimentally hypothesised states for adsorbed hydrogen on Pt, $H_{UPD}$ and $H_{OPD}$, revealing their role in governing interfacial water reorientation and hydrogen evolution. The transition between these two H states with increasing potential, induces a shift from a hydrophobic to a hydrophilic interface and correlates with a change in the primary electrode screening mechanism. This results in a slope change in differential capacitance, marking the onset of the experimentally observed peak around the potential of zero charge. Our work produces crucial insights for advancing electrocatalytic energy conversion, developing deep understanding of electrified interfaces.

The observable physical-chemical properties of electrified interfaces depend closely on their molecular structure. In turn, interfacial properties govern electrochemical processes, such as electrocatalysis and corrosion. Thus, understanding and controlling these properties at the molecular-level, is key to develop better catalysts. In particular, Platinum remains the catalyst of choice for many technologically significant reactions, such as the hydrogen evolution reaction (HER) in electrolysers. However, even for the ideally flat Pt(111)-electrolyte interface, key molecular-level details—structure, coverage, origin of wettability and capacitive response—remain largely unknown, both experimentally and theoretically.

This lack of insight reflects a broader challenge in electrochemistry: bridging the gap between macroscopic experimental observations and atomistic theoretical interpretations. Experimentally, the buried nature of active sites hinders a characterisation resolved at the molecular scale; theoretically, simplistic interfacial models or artificial assumptions about interfacial charge distribution, hinder accurate evaluation of key electrochemical parameters, including interface polarisation and the influence of the electrolyte on interface properties. Consequently, our knowledge of the fundamental physics and chemistry at electrified interfaces is fragmented, as further illustrated in the following sections for the Pt(111)-electrolyte interface.

The Pt-group metals have the ability to adsorb hydrogen at potentials higher than the standard potential for HER[1–4]. Two states of adsorbed hydrogen have been experimentally identified on Pt[3]: underpotential deposited hydrogen, $H_{UPD}$, more strongly adsorbed on fcc sites, observed at higher potentials, and overpotential deposited hydrogen, $H_{OPD}$, more weakly adsorbed on atop position at lower potentials.

The existence of underpotential deposited Hydrogen species, $H_{UPD}$, was initially suggested based on kinetic data for Pt(111) and polycrystalline platinum[1,5]. In acidic environment, underpotential deposition of hydrogen produces a characteristic adsorption signal in the cyclic voltammogram (CV), indicating that hydrogen starts adsorbing at around 0.4 V, and its coverage increases to the limiting value of 0.66 monolayers (ML) on Pt(111), as the RHE potential is approached[1,6]. However, other experimental evidence in similar conditions[7] suggests that $H_{UPD}$ adsorption would extend beyond this region and that coverage by adsorbed hydrogen on Pt(111) would be significantly larger than 0.66 ML, as also elaborated in theoretical studies[8,9]. Overpotential deposited hydrogen, $H_{OPD}$, starts adsorbing on atop sites slightly above the RHE, at potentials that compete with HER. $H_{OPD}$ is considered the active intermediate for the HER[1,5,10], while the role of $H_{UPD}$ in both HER and the hydrogen oxidation reaction (HOR) is more

[1]Department of Chemistry and Thomas Young Centre, Imperial College London, London, UK. [2]Department of Physics, Johannes Gutenberg University, Mainz, DE, Germany. [3]Department of Physics, Ruhr-University Bochum, Bochum, DE, Germany. [4]Present address: 4 rue Roland Oudot, Créteil, France. ✉e-mail: c.cucinotta@imperial.ac.uk

controversial. Some believe that $H_{UPD}$ is an intermediate for the HOR[10,11], highlighting the complexity of both HER and HOR, while others believe it is a spectator, only indirectly influencing the reactions by blocking surface sites[12]. This is a central issue in hydrogen electrochemistry as it impacts the understanding of the rates of technologically important reactions, such as HER in electrolysers and HOR in fuel cells.

Spectroscopy[13,14] and other evidence[15] supports the existence of two distinct H species on Pt showing that $H_{OPD}$ interacts with water molecules, while $H_{UPD}$ does not, and also confirm that $H_{UPD}$ adsorption occurs at fcc sites (inactive in IR spectroscopy), and $H_{OPD}$ adsorption at top sites (IR active)[5,16]. Unfortunately, the high Faradaic currents associated to HER/HOR may mask the adsorption/desorption currents, so it difficult to distinguish among different H adsorbed species in this potential range[7] or decouple the adsorption of hydrogen from that of other ions in solution. Thus, the specific interplay between the two adsorbed species as a function of the applied potential, their specific contribution to HER/HOR, and even the basic H coverage morphology of the Pt(111)-electrolyte interface are only qualitatively understood to date.

Another interesting open question is about the molecular origin of Pt wettability as a function of applied potential. Electrochemical Quartz Crystal Nanobalance (EQCN) analysis of Pt behaviour[2] reveals that around the potential marking the onset of $H_{OPD}$, the interaction between water and the electrode reaches a minimum, suggesting that adsorbed hydrogen species make the surface hydrophobic by altering surface dipole properties[17]; this picture is also supported by spectroscopy[5,18]. Unfortunately, no experimental technique can directly monitor in situ changes in the surface electronic and wetting properties of Pt as a function of electrode potential, so our knowledge of the specific interactions of these species among themselves, as well as with the electrode and water, remains, again, qualitative.

Recent ab initio molecular dynamics (AIMD) simulations have started to shed some light on the hydrogen adsorption behaviour on platinum surfaces in the presence of water electrolytes. Specifically, modelling the electrolyte with pure water, these simulations develop the concept that when hydrogen coverage on the platinum surface reaches 0.66 ML, the metal surface and water in the electrolyte can no longer facilitate charge redistribution due to adsorbed hydrogen and water molecules, leading to spontaneous hydrogen desorption[19]. These findings support the experimental hypothesis of a saturation H coverage close to 0.66 ML close to RHE potential, experimentally ascribed to the repulsive lateral interaction between adsorbed Hydrogen atoms[3,12]. However, although ions in the electrolytic solution have a central role in screening the electrode, they were not included in these AIMD simulations. Moreover, the two H adsorption states on Pt electrode were not identified. Further research is necessary to fully understand the complex behaviour of the two types of hydrogen adsorbed on the Pt(111) surface under bias.

A closely connected open issue also impacting energy conversion technology, is unravelling the molecular adsorption processes underlying the capacitive response of Pt(111)-electrolyte interface. In particular, above 0.4 V, in the double layer window where the Pt surface should be stable without adsorbates, the interface should behave as a Gouy–Chapman–Stern ideal double layer, and a minimum in differential capacitance corresponding to the potential of zero charge (PZC), should be measurable. Starting from Pajkossy and Kolb[20], capacitance measurements have been used to determine such minimum, which was eventually observed at 0.56 V only very recently, at pH 3 and very low ionic concentration (0.1 mmol)[4]; However, an unresolved issue is how to explain that the observed capacitance is higher than what is predicted with continuous theories such as the Gouy–Chapman–Stern model[4,21,22]. This anomalous behaviour could be partially rationalised in terms of the presence of weakly absorbed species over the surface, however, no consensus is achieved about the specific adsorption process. Some authors suggest that the anomalous behaviour is due to the potential adsorption of anions or cations[23], while others argue that it involves anions only[24] or attribute it to OH adsorption[25]. Although these theories can be semi-quantitatively reconciled[23], the exact nature of the ions and of their interaction with the Pt(111) substrate still remains a matter of

debate. In general, these studies all suggest that interfacial adsorption processes are key for determining the capacitive behaviour of Pt(111)-water interface. This notion is relevant at all electrolyte concentrations. Indeed, in addition to the minimum, a maximum in differential capacitance is also observed at most ionic concentrations near the PZC on Pt(111) electrode surfaces[4,20]. Initially, this maximum was associated with the PZC and attributed to reorientation of the water dipoles, but later it was excluded by Kolb[20,26] and more recently associated again with the PZC[4]. To interpret this capacitive behaviour close to the PZC, many concur that water adsorption at the interface is likely to play an important role[22]. In the past, Kornyshev[27] had explained for a different system that the capacitance peak corresponds to the voltage at which the potential drop across the electrical double layer (EDL) is equal to the thermal energy of the ions in the electrolyte solution. The negative slope in differential capacitance could be explained by the consideration that, above this potential, the ions in the electrolyte solution diffuse into the electrically charged surface, resulting in a decrease in the differential capacitance and electrode charge[24,27]. Other studies have shown that the application of a progressively more positive potential leads to an increase in coverage of positively charged water molecules chemisorbed over the surface[9,19,28,29]. The chemisorption dipole generated by these charged water molecules is mainly responsible for screening of the metal charge during polarisation[28,29]. Recent work by Cheng's group[29], modelling Helmholtz layers with very high ion concentration, shows that a change in water chemisorption in response to electrode potential leads to a bell-shaped differential capacitance around the PZC. However, this calculation models the limit of a very high interfacial ionic concentrations, which can alter the structure of the interface significantly. It does not consider how water and ions in solution can contribute to differential capacitance in a less constrained environment, or address how the presence of hydrogen on the surface affects water adsorption and re-orientation below the PZC. Thus, despite decades of research and these recent most intriguing findings, we still do not have a comprehensive understanding the molecular origins of differential capacitance close to the PZC, or its relation with adsorption processes over the surface, necessitating further analysis.

The above scenario highlights significant, interconnected questions about the molecular origins of the complex physico-chemical behaviour and morphology of metal/electrolyte interfaces. These include the origins of wettability, capacitance, the structure and interaction of adsorbed species, and the high catalytic activity of these materials. Current theoretical and computational models have not been able to date to provide comprehensive molecular picture of this complexity.

Advancing our understanding requires the development of more realistic, unbiased computational models that simulate well-defined conditions, which can help to interpret and contrast experimental evidence resulting from varied or uncontrolled conditions.

In this paper, we develop highly realistic and controlled AIMD models for the hydrogen-covered Pt/water interface, representing well-defined states with specific electrode coverage, temperature and charge, across a range of potentials from negative up to PZC. Our unique models provide a holistic perspective into the complex interplay between the aqueous electrolyte, adsorbed hydrogen and the electrode, uncovering molecular-level correlations between interface hydrophilicity, hydrogen coverage, catalytic activity and capacitive response as the potential changes.

Understanding the molecular-level electrochemistry of metal/water interfaces and in particular, H adsorption behaviour is essential for advancing hydrogen electrochemistry and determining energy conversion rates in electrolysers and fuel cells. This knowledge is also crucial for corrosion prevention and enhancing supercapacitor technologies, impacting both scientific research and industrial applications.

## Results

We systematically investigated the impact of electrode potential on the molecular structure, Hydrogen coverage, polarisability and wettability of the Pt-aqueous solution interface. To achieve this, we constructed a series of models for the electrified Pt(111)-water solution interfaces (see Fig. 1a),

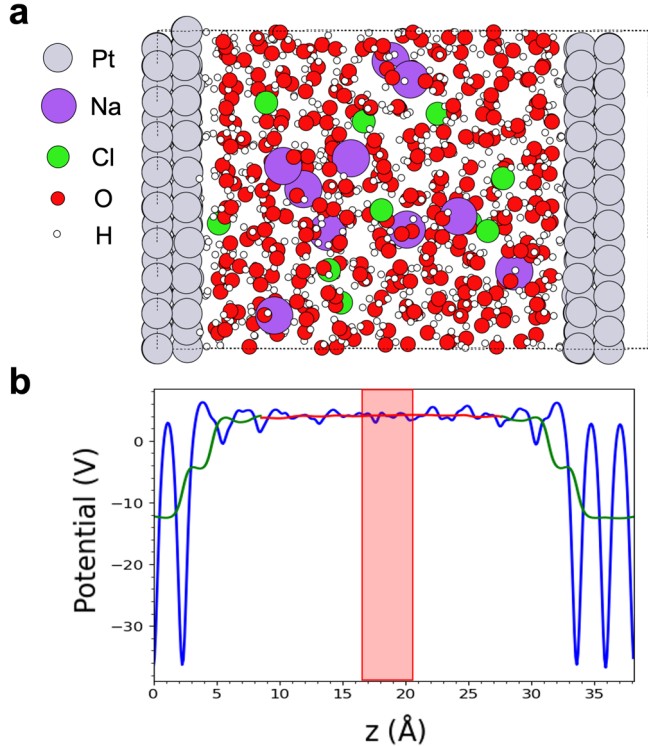

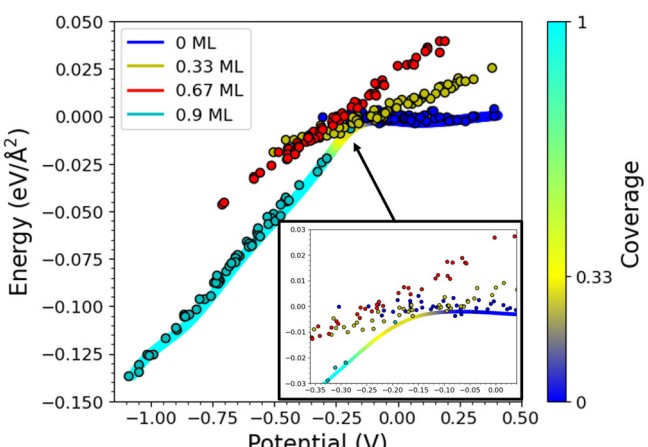

**Fig. 1 | Model structure and potential profiles for the Pt/ water electrolyte interface. a** Structure of the Pt/electrolyte interface. **b** Profiles for the average potential (blue line), double mean potential for the electrode (green line) and the double mean potential for the water bulk (red line). The red box indicates the window where the water bulk potential was calculated.

**Fig. 2 | Hydrogen adsorption energies and surface coverage as functions of electrode potential.** The zero of the potential scale is aligned to the potential of zero charge (PZC) of the platinum-water electrolyte interface. Coloured dots depict snapshots from simulations at constant hydrogen coverage, as labelled. The inset illustrates how the distribution of configurations changes as the potential shifts from the PZC towards more reducing values. The solid coloured lines display the average re-weighted formation energy (in eV/Å²) at each potential, calculated as detailed in the Methods section. The colour bar on the right provides the colour coding for the different coverage levels.

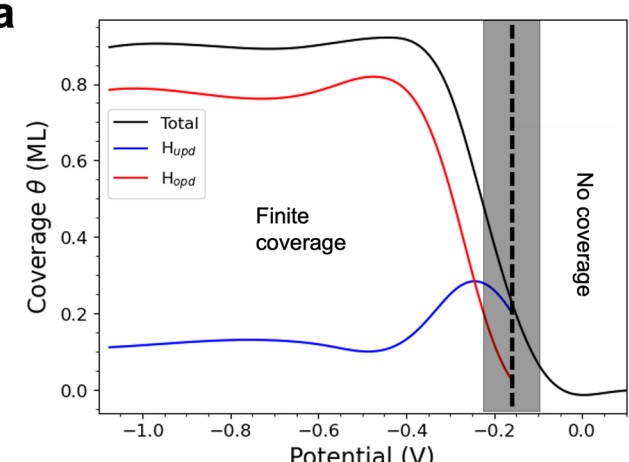

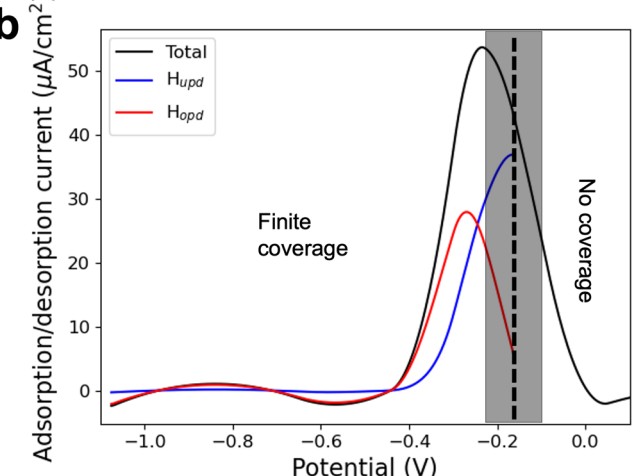

**Fig. 3 | Coverage and adsorption/desorption current as a function of the applied potential. a** Coverage ($\theta$) as a function of the applied potential. The overall coverage is depicted in black, while the contributions of $H_{UPD}$ and $H_{OPD}$ are indicated in blue and red, respectively. **b** Adsorption/desorption current $I$, ($I \propto \theta$) as a function of the applied potential. The total current is plotted in black, while the $H_{UPD}$ and $H_{OPD}$ contributions are in blue and red. The shaded boxes indicate the confidence range for the transition between finite and zero coverage.

window, as our focus is on understanding how the surface is covered with hydrogen at potentials consistent with the H evolution reaction, and slightly above. The Materials and Methods section provides a detailed account of the models and the statistical approach used to evaluate the expectation values for the formation energy and coverage as a function of the potential, as well as how the interfacial potential drops have been used to align the electrode potentials to the PZC.

### Hydrogen coverage as a function of the applied potential

Our first goal has been to determine the relation between equilibrium hydrogen coverage on Pt(111) and applied potential and to characterise the structure of hydrogen adsorption. Figures 2 and 3a show our analysis of the hydrogen adsorption enthalpy and coverage as functions of the applied potential.

Our findings align well with experimental data, indicating that hydrogen coverage begins at ~0.12 V below the PZC. From this potential, hydrogen coverage increases progressively as the potential decreases, reaching about 0.9 ML at around 0.4 V below the PZC.

We reference our potential scale to a flat Pt(111)/water electrolyte configuration at the PZC, under pH conditions with no adsorbed species on the surface, but with a 2M concentration of Na⁺ and Cl⁻ ions in solution. It is important to note that the PZC can vary due to environmental factors such

hydrogenated with varying levels of surface H coverage, including low (~0.3 ML), medium (~0.6 ML) and high (~0.9 ML) coverage.

These models simulate a potential window ranging from −1.0 to 0.5 V relative to the potential of zero charge (PZC), where the Pt surface is known to be either clean or covered by H. In this paper, we have not considered the specific adsorption of OH⁻, expected at potentials above the double layer

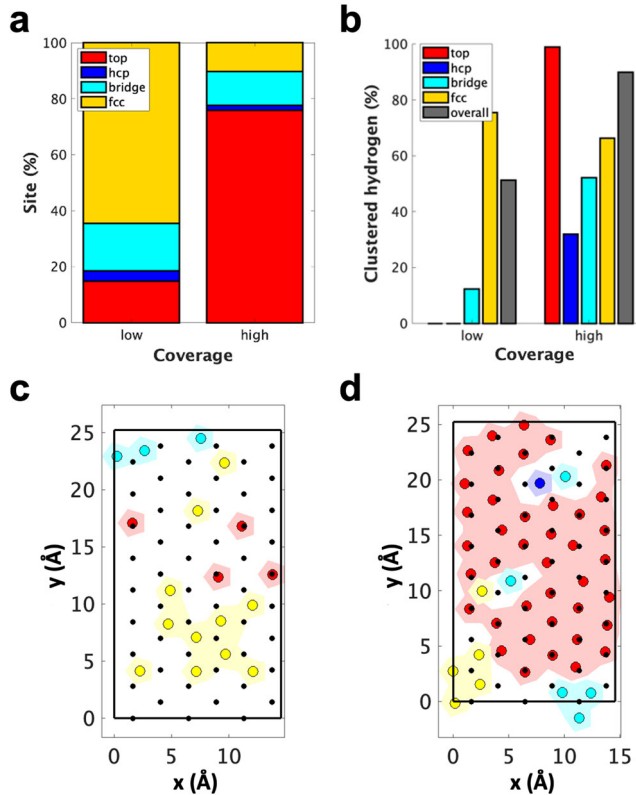

**Fig. 4 | Distribution of adsorbed hydrogen on Pt at low and high coverages.**
**a** Distribution of adsorbed hydrogen on Pt at low and high coverages. **b** Percentage of hydrogen atoms belonging to a cluster. **c** Schematic representation of adsorbed H distribution on a low-coverage surface. **d** Schematic representation of adsorbed H distribution on a high-coverage surface. In **c** and **d**, the shaded areas indicate H clusters. The small black dots represent Pt surface atoms, while the larger circles denote adsorbed hydrogen. The colour coding corresponds to the sites indicated in (**a**) and (**b**).

as pH, dielectric constant, surface structure, electrolyte nature, and pressure, making it difficult a direct comparison with many experiments. The $H_2$ reversible potential also depends on these factors and thus affects the onset of $H_{OPD}$ deposition. However, this variability does not impact the range of $H_{UPD}$ relative to the onset of $H_{OPD}$, as both processes follow the same pH dependence[10].

A significant observation, depicted in Fig. 4, is the clustering of hydrogen atoms on the Pt surface. This challenges the conventional understanding, based on experimental suggestions and prior simulations in vacuum, which proposed a uniform distribution of H atoms on Pt(111) surfaces due to their mutual repulsion. Instead, we observe a high degree of heterogeneity in the distribution of adsorbed hydrogen atoms as the potential varies. This behaviour can be attributed to the crucial role played by the electrolytic solution in shaping the adsorption behaviour of hydrogen on metal surfaces (Fig. 4c, d and Supplementary Fig. S1). A more detailed explanation of this clustering, in terms of hydrogen charges and cluster adsorption energy, is given in Supplementary Fig. S11 and Supplementary Discussion 3, 'Thermodynamic origin of cluster stabilisation'.

### Identification of $H_{UPD}$ and $H_{OPD}$
Through the analysis of the predominant adsorption structure at each coverage, we identified computationally $H_{UPD}$ and $H_{OPD}$, the two experimentally hypothesised states for adsorbed hydrogen. To the best of our knowledge, this is the first direct AIMD observation of the dynamic interplay between these species under varying potential. It sheds new light on their role in determining water arrangement, driving the electrochemical behaviour of the interface, and facilitating HER/HOR.

When varying the applied voltage, we observe two different hydrogen-adsorbing species. Hydrogen identified as $H_{UPD}$ starts adsorbing on fcc sites, with coverage increasing until it peaks around −0.3 V versus the PZC. Below −0.3 V versus PZC, the predominant adsorption site becomes atop, and we identify the H atoms adsorbed there as $H_{OPD}$. As the potential decreases, $H_{OPD}$ coverage rapidly increases while $H_{UPD}$ clusters shrink (Fig. 3a and Supplementary Fig. S2). The potential region between the point when $H_{OPD}$ becomes dominant and when H coverage becomes maximum marks the onset of HER. This trend continues until a maximum overall coverage of 0.9 ML is achieved at ~−0.4 V versus the PZC.

Notably, our deconvolution of the coverage curve into $H_{UPD}$ and $H_{OPD}$ components matches experimental evaluations of strongly and weakly adsorbed H as shown by Chen[11] and by Strmčnik[7]. The molecular-level picture of the interaction between $H_{UPD}$ and $H_{OPD}$ clarifies several experimentally debated questions. Firstly, H continues to adsorb on fcc sites even below the potential for HER, traditionally identified as the limit for underpotentially deposited H[5]. Secondly, the overall surface coverage can exceed 0.66 ML, aligning with experimental observations that true hydrogen coverage on Pt(111) is significantly larger 0.66[5]. Lastly, despite the dominant adsorption site changing with potential, we always observe a mix of both types of sites and a minimal proportion of bridge and HCP sites, indicating the coexistence of both $H_{UPD}$ and $H_{OPD}$.

A more detailed analysis of H distribution in prototypical low (0.3 ML at −0.2 V versus PZC) and high (0.9 ML at −0.6 V versus PZC) coverage configurations shows that at high potentials and low coverages, more than 60% of the adsorbed atoms occupy fcc sites, forming $H_{UPD}$ clusters that cover ~20% of the metal surface. The remaining H atoms are predominantly adsorbed at isolated bridge or atop sites (Fig. 4c). When the overall coverage is ~0.9 ML, the majority of the adsorbed atoms occupy atop sites, forming clusters that cover ~60% of the metal surface (Fig. 4a, d).

We would like to emphasise that, unlike in Todorova's work[19], our simulations did not show an overall H coverage saturation at 0.66 ML. Currently, we observe saturation only for $H_{UPD}$ fcc coverage. Indeed, the 0.66 ML structure was found to be metastable at all potentials, with only a few stability points in the region at the boundary between $H_{OPD}$ and $H_{UPD}$ predominance regions. Any attempts to increase its size above ~60% adsorbing H over fcc sites, led to H desorption into the solution phase, indicating the instability of this configuration. However, below the onset of $H_{OPD}$ region, we observed that the overall coverage can actually become higher than 0.66 ML, by adsorbing H on atop sites. This is in line with the experimental observation that hydrogen coverage on Pt(111) can be significantly larger than 0.66 ML below the $H_{OPD}$ onset potential[7].

Notably, our ability to observe an H coverage higher than 0.6 ML is likely due to the inclusion in our model of ions in solution, which can contribute to the screening of the highly negative surface charge at low potentials.

We also evaluated the current associated with the variation of adsorbed hydrogen species on the surface as a function of the potential, $I = \pm \nu Q_{tot} \frac{d\theta}{dV}$, in the limit of a slow scanning rate $\nu$, assumed to be 50 mV/s. Here, we used a slightly modified version of Karlberg's approach[8], which accounts for charge polarisation of the double layer under applied potential (Fig. 3b) (see 'Methods' section). To compare our findings with experiments, we need to clarify that our first-principles model does not include current from hydrogen evolution or anion adsorption. Comparison with the experimental evaluation of this component of the hydrogen peak in CV curves for Pt(111)[12] shows that we are able to replicate the experimental data very closely. Notably, we have also de-convoluted the current in its $H_{UPD}$ and $H_{OPD}$ components.

The coexistence and interplay of $H_{UPD}$ fcc and $H_{OPD}$ atop clusters have significant implications for the electrocatalytic activity of the electrode, especially in the context of the HER. In particular, since in reducing conditions the size of the $H_{OPD}$ cluster increases when the potential decreases, there is an increasingly high probability that HER involves $H_{OPD}$ as the potential becomes more negative.

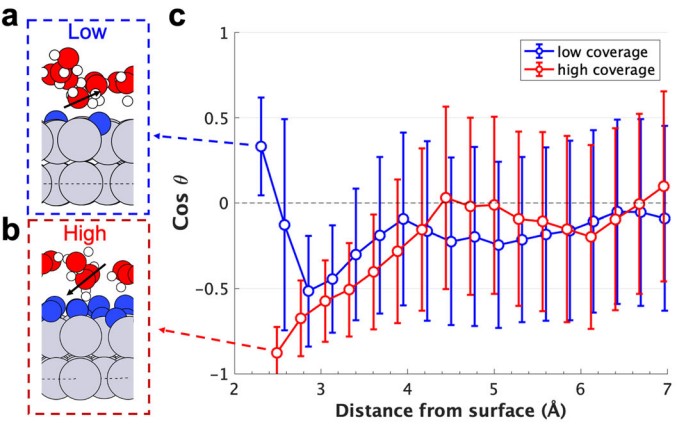

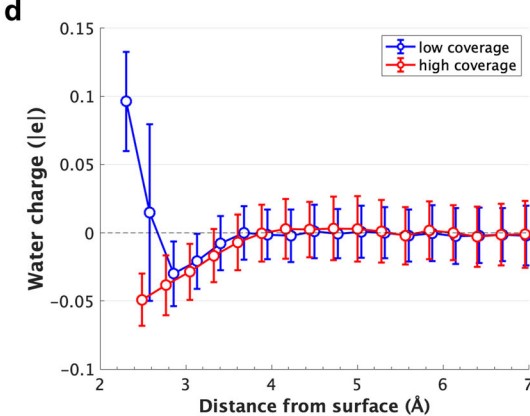

**Fig. 5 | Arrangement and charge of water on the Pt surface. a** Position of a water molecule in the first water layer on a low coverage surface. **b** Position of a water molecule in the first water layer on a high coverage surface. **c** Average cosine of the angle between the water dipole and $z$-axis as a function of the distance from the surface. **d** Average charge of the water molecules as a function of the distance from the surface for low and high coverage surfaces.

As detailed in the next paragraph, at low potentials, when HER takes place and the percentage of $H_{UPD}$ drastically reduces, the approach of a water molecule or hydronium ion with a hydrogen pointing towards the surface is more favourable. Consequently, HER can efficiently proceed in this potential range.

Although the primary objective of this paper is unravelling the equilibrium distribution of hydrogen across the surface as a function of potential rather than the mechanism for HER, the experimental assertion proposing the involvement of two distinct hydrogen species in HER[14] is consistent with our findings, given the presence of both species on the surface. Moreover, the observation of few exchange events between $H_{UPD}$ and $H_{OPD}$ during dynamics may support the idea of an exchange energy difference comparable to thermal equilibrium. This would provide a positive answer to the longstanding question of whether $H_{UPD}$ is involved in HER[7,10–12]. At high potentials, where the $H_{UPD}$ fcc cluster is predominant, the first layer of water orients its O atom towards the surface, which makes proton transfer to the surface more difficult.

## Hydrophilicity-hydrophobicity modulation induced by H coverage

The analysis of our AIMD trajectories reveals that density, charge, and orientation of interfacial water are strongly influenced by H coverage and therefore vary significantly with the electrode potential.

In particular, analysing Figs. 5 and 6 as well as Supplementary Figs. S4–S9 and Supplementary Discussion 2), we can see that when no H is present over the Pt surface, and until H coverage is low, water can approach the surface, chemisorbs almost flat over it, with its H atoms pointing slightly outwards, is positively charged, and has coverage which decreases with potential, consistently with other literature[9,28,29], and further decreases when H is present. Interestingly, the Pt bonded to chemisorbed water is also positive and the charge donated by water distributes over the neighbour Pt atoms, which become more negative. After $H_{OPD}$ start adsorbing and H coverage increases to 0.67 ML, the adsorbed layer prevents a direct interaction between the oxygen atoms from water and the negative electrode surface. As a result, a chemisorbed water layer cannot form, the closest water molecules are pushed away from the surface and point one of their positive hydrogen atoms towards the negative surface. Overall, this water layer is charged slightly negatively and its orientation closely resembles the one found for the second water layer on a clean Pt(111)-water interface[28] and on highly hydrophobic graphene surfaces[30] (Figs. 5, 6 and Supplementary Figs. S4–S6). Cicero[31] showed that a metal surface displaying the low coverage structure exhibits hydrophilic behaviour due to the stronger hydrogen bonds established between the water bonded to the surface and the surrounding water molecules (Supplementary Figs. S7, S8). In summary, the surface hydrophilicity is modulated from being strongly hydrophilic to

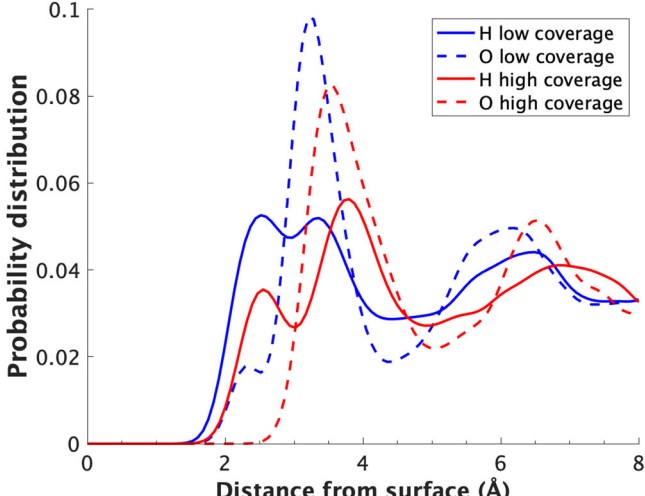

**Fig. 6 | Hydrogen and oxygen probability distributions.** Hydrogen (solid lines) and oxygen (dashed lines) probability distributions for low (blue) and high (red) hydrogen coverage surfaces as a function of distance from the surface.

weakly hydrophobic depending on the applied potential. This transition is induced by the changes in surface H coverage, which, in turn, substantially affect surface charge distribution and water orientation.

Notably, the main features of the interfacial structure for high coverage, are also observed for the structure simulating the metastable 0.67 ML coverage, as shown Supplementary Information file, in Figs. S4–S9, therefore the transition between hydrophilic to partially hydrophobic regimes approximately occurs below −0.3 V wrt. the PZC, just below the potential when $H_{OPD}$ becomes predominant.

In line with the above findings, the number of available acceptors among water molecules progressively drops to zero as the potential decreases, reaching zero between the 0.33 ML and 0.67 ML coverage regions. The number of hydrogen donors also decreases as hydrogen coverage increases (see Figs. S7 and S8 in the Supplementary Information document). This is consistent with a decreasing number of chemisorbed water molecules connecting via their hydrogen atoms with other water molecules (Figs. S4–S6 in the Supplementary) as the potential decreases.

Our findings match the experimental evidence of hydrophobicity of Pt(111) induced at negative potentials by H deposition[17,18]. They also align with recent experimental findings[32] which use infra-red spectroscopy to explore the interactions between adsorbed hydrogen and water and show that when the potential decreases and H coverage becomes high, hydrogen

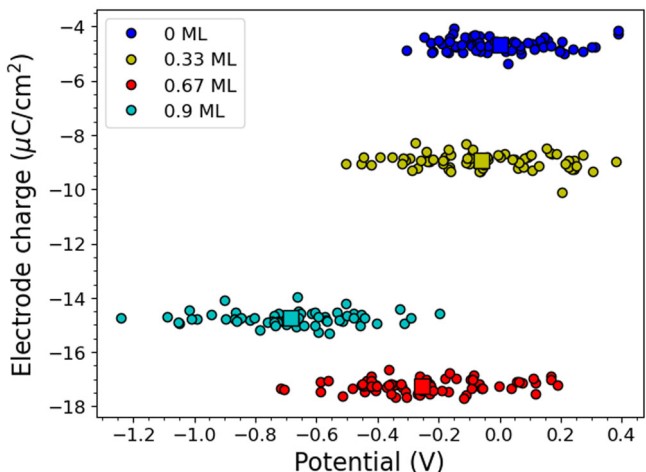

**Fig. 7 | Surface charge versus applied potential.** Electrode surface charge, measured in units of |e| for 0, 0.33, 0.66 and 0.9 ML coverages as function of the potential. The circular markers indicate the charge of individual snapshots while the square marker represents the mean value for each coverage.

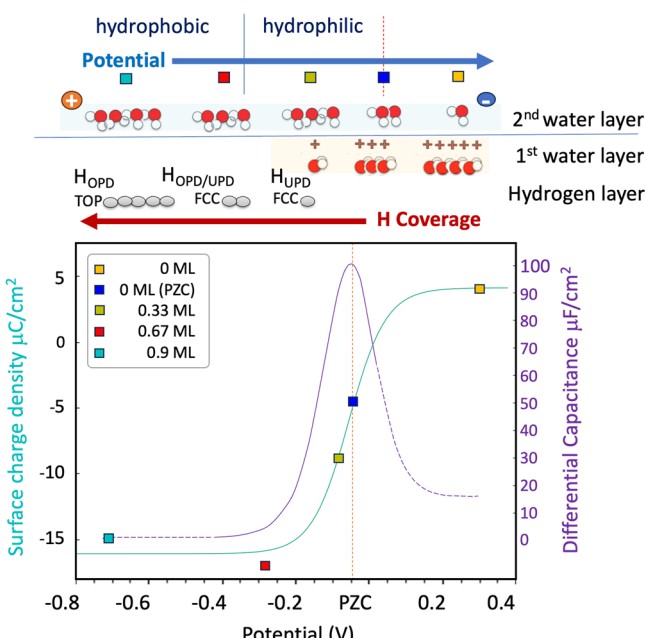

**Fig. 8 | Surface charge density and differential capacitance versus applied potential.** Teal line: surface charge density ($\sigma$, measured in $\mu C/cm^2$) vs. applied potential in the potential region where changes in water orientation and screening mechanism of the electrode occur. Purple line: differential capacitance (C, in ($\mu F/cm^2$)) in the same potential range. At the top, pictorial representation of water orientation and coverage as well as Hydrogen coverage. The orange dotted vertical lines, signal the potential of zero charge. The blue dotted vertical line represents the boundary between hydrophobic and hydrophilic domain. The dashed regions in the differential capacitance curve are plotted to guide the eye and have not been evaluated in current work. They point to 0 on the right and to 20 $\mu F/cm^2$ as per ref. [28].

appears to displace water molecules. Our data are also consistent with the EQCM experimental observation[2] that a minimum of the interaction between water and Pt substrate is reached at the minimum of water coverage, which occurs just below the onset of $H_{UPD}$ region.

This picture is confirmed by the analysis of the mass density distribution for low coverage, where two distinct oxygen peaks are present below 3.8 Å (dashed lines in Fig. 6 and Supplementary Fig. S6), as for clean interfaces[9,19,28,29,31]. The first small peak, located at 2.0–2.7 Å from the surface, characterises the position of the first chemisorbed water layer which chemisorbs almost flat over the surface. This water configuration is induced by the formation of a dative-like bond between one of the water molecule's lone pairs and the metal surface[19]. This bond causes the two hydrogen atoms and the remaining lone pair to lie on the same plane slightly above the oxygen atom, as shown in Fig. 5a. The chemisorbed nature of the first water layer is confirmed by the observation that in the time spanned by our trajectories for low and no coverage systems, when a water molecule forms such a bond, its diffusivity is heavily impaired. In addition, a localisation of electronic charge along the Pt-water bond is also observed, as shown in a sample snapshot in Supplementary Fig. S9.

The second water layer arises at 2.7–3.5 Å from the surface, however, the dative bond does not form in this case, and the water molecules become slightly negative, oriented with one of the hydrogen atoms toward the negative platinum atoms of the surface, and the other pointing slightly away, similarly to the water configuration when there is a high H coverage at negative potentials (Fig. 6 and Supplementary Figs. S4–S6). This second water layer is more free to reorient as a response to the potential, however, its intrinsic dipole is smaller than that characterising chemisorbed water[9,29]. Beyond 3.8 Å, the probability distribution flatten out (see Supplementary Fig. S5) and the overall water charge is zero (see Fig. 5d), as the interaction with the surface is very small and water bulk conditions are approached.

**Interfacial capacitance**
We have illustrated previously that the variations in hydrogen coverage with potential, play a crucial role in influencing the charge, orientation, and density of water. This ultimately affects the interface capacitance. It has been shown previously that at very low or very high potentials, the differential capacitance saturates towards a constant value[29]. We focus here on the potential range going from sligthly negative potentials, up to the PZC.

The electrode charge of the bare, low, medium, and high coverage structures are shown as a function of potential in Fig. 7. The graph shows that the bare surface has an overall negative average charge, amounting to $-9.2\,\mu C/cm^2$ ($-2.1$ |e|). The low coverage surface follows with

$-18.0\,\mu C/cm^2$ ($-4.1$ |e|) while the high coverage surface has $-29.8\,\mu C/cm^2$ ($-6.8$ |e|). The ~0.6 ML coverage configuration has only a couple of stability points, and overall is metastable, with a charge of about $-34.2\,\mu C/cm^2$ ($-7.8$ |e|), however, we use this configuration to evaluate the capacitance at the boundary between hydrophobic and hydrophilic regions. Notably, this configuration is more negative than 0.9 ML configuration, which is more stable at lower potentials. The lower charge in the ~0.6 ML configuration is the result of the occurrence of more $H^+$ desorption processes than in 0.9 ML case during the equilibration phase, which left the metal slab more negatively charged. This enhanced desorption tendency is due to the metastable nature of the fcc cluster above 0.6 ML coverage[19]. For the point at V = +0.4 V vs. PZC, we have used the data from our previous work on Pt(111)-water interface around the PZC, calculated with a consistent set of methodologies and parameters[28].

The plot of surface charge versus potential (see Fig. 8) exhibits a sigmoidal trend, similar to an electrosorption isotherm for charged species over a surface. To understand this we need to recall that, as demonstrated earlier, when the potential becomes higher than ~$-0.3$ V, the system transitions from hydrophobic to hydrophilic, due to the decreasing H coverage which starts allowing the chemisorption of water molecules which are charged. This transformation is clearly reflected in the observed change of slope in the charge-versus-potential curve (Fig. 8). Notably, this corresponds to the onset of a peak in differential capacitance, obtained by taking the derivative of the charge/potential sigmoidal curve, which reaches its maximum around the PZC (Fig. 8).

More specifically, at very negative potentials, when the metal surface is saturated with adsorbed hydrogen, the capacitance remains constant, as previously noted in the literature[33–38]. In this potential range, the interface is hydrophobic (even if it must allow water or hydronium ions to approach for H evolution) and lacks of a chemisorbed water layer. Therefore the electrode charge is screened by solvated ions and water molecules, orienting their H

atoms towards the electrode surface (Fig. 5 and Supplementary Fig. S4). Above ~−0.3 V, when the interface transitions from hydrophobic to hydrophilic, the primary electrode screening mechanism changes. In this potential range, extending up to the PZC, the first water layer reorients into an almost flat configuration, acquires a positive charge, is closer to the metal surface and chemisorbs onto it. Notably, the number of positively charged water molecules chemisorbed over the surface, increases with potential, resulting in a steep increase in differential capacitance. The steeper slope is explained as the water chemisorption dipole is larger than the water reorientation dipole. Thus, water chemisorption dipole becomes the main contributor to electrode screening and surface capacitive response[19,28,29].

The maximum capacitance is reached around the PZC, consistently with previous findings[29]. This aligns with the shared view[19,28,29] that water in the double layer region has peculiar properties: can carry charge, chemisorb onto surfaces, and screen electrode charges by generating a chemisorption dipole.

While we did not specifically focus on determining the capacitive response of the interface in the region above the PZC, we suggest that as we cross the PZC, the interplay between increasing the coverage of chemisorbed water molecules, and the counteracting action of the second water layer and different ions in solution, induces a flex in the slope of the surface charge vs. potential curve, identifying the maximum in the peak in differential capacitance.

Previous studies[29] observed that as the potential continues to increase above the PZC, water coverage saturates, while the differential capacitance starts decreasing until it reaches a constant value. Thus, when water coverage saturates, the main contributors to the variation in differential capacitance would become the other ions in solution, including $OH^-$ possibly chemisorbed over the metal surface and contributing negatively to differential capacitance.

Similarly to Cheng's findings for the case of very high ion concentration[29], we find a maximum in differential capacitance associated with the PZC; however it is important to note that, unlike in Cheng's work, our PZC doesn't signal the moment when water reorients from adsorbed and H-down to chemisorbed and flat. In Cheng's work, such reorientation is triggered by the densely populated ions of opposite nature present in the Helmholtz layer a the two sides of the PZC. In our study, not constrained by high ion concentration conditions, water reorientation occurs when the surface switches from hydrophobic to hydrophilic, ~0.2 V below the PZC. This transition marks the onset of the capacitance peak, which we've observed to be sharper than in Cheng's findings, consistently with experimental observation.

Similarly to Koper's findings[23], our study explains the molecular origin of differential capacitance around the PZC with the adsorption and interplay of ionic species over the surface. However, in our work, we demonstrate that the chemisorbed, positively charged species mostly responsible for the capacitive response just below the PZC are water molecules behaving as ions, distinct from the ions in the electrolyte, while the interplay of chemisorbed H and water layer is crucial to determine the physicochemical properties of the interface at lower potentials.

## Conclusions

By developing highly realistic AIMD models for the electrified Pt(111)-water-electrolyte interface at prototypic coverages (0, 0.3, 0.6 and 0.9 ML), we shed light on the complex interplay between the aqueous electrolyte and adsorbed hydrogen as the applied potential changes. We demonstrate a holistic connection between variations in H coverage, interface hydrophilicity, catalytic activity, and capacitive response, across reducing potentials, up to the potential of zero charge.

Our simulations reveal a bimodal H adsorption behaviour on Pt(111) in response to electrode potential variations. Adsorbed H atoms tend to cluster on the metal surface and form multiple domains with different adsorption sites.

Hydrogen starts adsorbing on Pt(111) at ~0.12 V below the PZC. From this potential, hydrogen coverage progressively increases as the potential decreases, reaching about 0.9 ML at around 0.4 V below the PZC.

Through the analysis of the predominant adsorption sites at various electrode potentials, we computationally identify the two experimentally hypothesised hydrogen adsorbing species: $H_{UPD}$ and $H_{OPD}$. $H_{UPD}$ adsorbs on fcc sites and is predominant at low hydrogen coverage (<0.3 ML) and high potentials. Below −0.3 V versus the PZC, $H_{OPD}$, which adsorbs on atop sites, becomes predominant, and hydrogen coverage rapidly increases to ~0.9 ML.

We clarify several experimental debated issues: firstly, hydrogen continues to adsorb on fcc sites even below the potential for HER, traditionally the limit for underpotentially deposited hydrogen. Secondly, the overall surface coverage can exceed 0.66 ML, aligning with observations that hydrogen coverage on Pt(111) can be significantly larger than 0.66 ML. Lastly, despite the dominant adsorption site changing with potential, a mix of both types of sites is always observed, with minimal bridge and hcp sites, indicating the coexistence of both $H_{UPD}$ and $H_{OPD}$.

We evaluate the adsorption/desorption current, which compares well with experiments, and uncover the mechanism of transition between these two states as the potential increases and their significant impact on interfacial water charge, polarisation and reorientation.

We demonstrate that this transition modulates a change at the interface from weakly hydrophobic to strongly hydrophilic, correlating with changes in the primary electrode screening mechanism and the onset of the bell-shaped peak in differential capacitance.

Specifically, at negative potentials, when the surface is saturated with adsorbed hydrogen, the interface displays predominantly hydrophobic behaviour. In this state, water molecules orient with their hydrogen atoms facing the electrode surface, and a chemisorbed water layer is absent. Recent literature[29] indicates that the differential capacitance remains constant in this potential range. Around −0.3V wrt. the PZC, the interface undergoes a transition from a hydrophobic to strongly hydrophilic, driven by the changes in the H coverage structure, which presents lower coverage. In this state, extending up to above the PZC, the first water layer reorients into a flat configuration, acquires a positive charge, moves closer to the metal surface, and chemisorbs over the metal. Notably, the coverage of these chemisorbed and charged water molecules increases with the potential. The associated chemisorption dipole, which is larger than the reorientation dipole, causes the charge density to steeply increase with potential. This sharper increase reflects a change in the primary electrode screening mechanism and manifests in the onset of the bell-shaped peak in differential capacitance.

As the potential passes through the PZC, a peak in differential capacitance is identified, in line with prior research[29]. However, it is important to note that, in our case, the PZC does not correspond to water reorientation from H-down to chemisorbed and flat, as reported by Cheng for systems with a very high ion concentration. In our investigation, water reorientation rather occurs at the boundary between the $H_{OPD}$ and $H_{UPD}$ regions, and coincides with the surface's transition from hydrophobic to hydrophilic. Furthermore, water reorientation doesn't mark the peak in differential capacitance, but rather marks its onset. Notably, we observe the differential capacitance peak to be sharper than in Cheng's research, in line with existing experimental data.

Our study links the peak in differential capacitance to the adsorption and interaction of different ionic species on the surface. While we did not focus into determining the capacitive response of the interface in the region above the PZC, we suggest that as we cross the PZC, the interplay between increasing the coverage of chemisorbed water molecules, and the counteracting action of the second water layer and solvated ions, induces a flex in the slope of the surface charge vs. potential function, identifying a maximum in the peak in differential capacitance. This effect may be further influenced by the chemisorption of $OH^-$ onto the surface, potentially contributing negatively, along with solution ions, to the capacitive response of the surface at higher potentials.

The interplay between $H_{UPD}$ and $H_{OPD}$ also affects the electrode's electrocatalytic activity, especially the HER. Indeed, at high potentials, water reorientation induced by the presence of H hinders proton transfer, but at lower potentials, when HER occurs, water orientation favours proton transfer, enabling efficient HER. We observe that for each coverage, the size of the predominant cluster increases when the potential decreases, therefore although both $H_{UPD}$ and $H_{OPD}$ are present at negative potentials, there is an increasingly high probability that HER involves $H_{OPD}$ as the potential becomes more reductive.

Overall, our simulations show that understanding the complex interplay between the interfacial water charge, H adsorption configuration, and potential is crucial for controlling interface behaviour. This has significant implications for practical applications in electrocatalysis, corrosion resistance, and energy storage devices. Applying our methodological approach to various materials, interfaces, and environmental conditions will disclose new knowledge in electrochemistry. For instance, it will help to identify active sites in their environment and reaction mechanisms for electro-catalytic transformations and how they depend on electrochemical conditions (e.g. type of ions in the electrolyte).

Despite the increased realism of the proposed AIMD models, we are still far from simulating EC systems from atom to device, due to necessary simplifications arising from computational costs. A significant limitation of AIMD is the short sampling time, restricting, for instance, the simulation of phenomena like ion diffusion, which occurs over much longer timescales. Additionally, achieving convergence for the Galvani potential drop becomes challenging when the Debye length of the electrolytic solution increases (at lower ionic concentrations). Machine learning-based interatomic force fields for electrified interfaces[39–41], developed from DFT datasets, offer promise due to their low computational overhead, though they face challenges in treating the long-range components of interatomic interactions.

Another inherent limitation is that controlling the potential is achieved by controlling the charge over the electrode at a constant number of particles. In it cumbersome to control simulations cumbersome and evaluate electrochemical reaction kinetics non-trivial. Therefore, developing grand-canonical methodologies to model electrochemical interfaces seems a natural solution, which is being pursued by an increasing number of researchers[42–45].

## Methods

We generated highly controlled datasets from AIMD trajectories that sample the Pt(111)-water-electrolyte interface at a wide range of applied potentials. These datasets provide statistically meaningful microscopic sampling of defined states with specific electrode coverage (0, 0.3, 0.6 and 0.9 ML), temperature, charge, and potential. This enables us to classify and reweight the database of states accumulated at each potential and extract physicochemical insights from each individual prototypical trajectory. In the following paragraphs we describe the computational setup for these simulations, the method we use for establishing a potential scale and the statistical approach to generate the expectation value for energy and coverage as a function of the potential.

### Computational setup

Calculations were performed within the density functional theory (DFT) using CP2K package[46]. The PBE[47] formulation of the general gradient approximation and Tether, Goedecker and Hutter (GTH) pseudopotentials[48,49] were used while Van der Waals interactions were accounted under the DFT-D3 scheme.[50] A cutoff energy of 300 Ry was applied to describe the electronic density. Only the $\Gamma$-point was included in the Brillouin zone (BZ) integration. AIMD simulations were carried out with Car-Parrinello-like (CP-like) MD with a 0.5 fs timestep[51]. Systems were sampled with a Langevin-type equation governed by a random-noise term, $\gamma_D$, which is highly system-dependent. We indicated the employed parameters for each simulated structure in Table 1. The $\gamma_D$ term allowed to compensate for the intrinsic dissipation of the Car-Parrinello-like methodology and to keep the temperature between 326 and 331 K (see Table 1 for

**Table 1 | Simulation details regarding the computed structures: random-noise term $\gamma_D$, average temperature, and length of the cell in the z-direction**

| Coverage (ML) | $\gamma_D$ | Temperature (K) | Cell length (Å) |
|---|---|---|---|
| 0 | 2.9 e$^{-4}$ | 331 | 36.9 |
| 0.33 | 2.8 e$^{-4}$ | 326 | 38.1 |
| 0.67 | 2.8 e$^{-4}$ | 336 | 39.3 |
| 0.9 | 3.1 e$^{-4}$ | 326 | 38.9 |

details). In CP-like MD forces are not obtained by full minimisation of the wave-function as in Born-Oppenheimer MD. Instead, they follow a predictor-corrector scheme based on the previous steps wave-functions. An extrapolation order of 1 for the always stable predictor-corrector was used. The number of corrector steps was determined either by the fulfilment of a convergence threshold ($5.10^{-5}$ a.u.) or by a maximum cap of iterations[51]. The corrector stepsize was set to 0.0075.

The trajectories were equilibrated for 80 ps because of the slow mobility of the ion dissolved in water and to stabilise the systems after the occurrence of several H desorption processes from the surface (see below), and analysed for about 30 ps each. All the simulated systems are composed of two symmetrical and independent half-cells. The metal slab (shown in Fig. 1a) was modelled using a 4-layer Pt electrode exposing the (111) surface to the saltwater (NaCl). The coordinates of the central layers of the Pt slab were constrained to the bulk electrode values. Overall, there are 216 Pt atoms, 330 water molecules, ten couples of Na$^+$ Cl$^-$ ions, corresponding to ~2 M NaCl concentration, and a variable number of hydrogen atoms adsorbed on the surface. Namely, we introduced an initial coverage of 0 ML (bare surface), 0.33 ML (referred as low coverage surface), 0.66 ML (referred as medium coverage surface) and 1 ML (referred as high coverage surface).

During the equilibration phase of the dynamics, we observed notable events, where a few hydrogen atoms desorbed from the platinum surface and entered the solution phase, resulting in a slight decrease in the initial coverage for each configuration. Specifically, 5 out of 36 hydrogen atoms desorbed from the low-coverage surface, 9 out of 72 for the medium coverage, and 8 out of 108 for the high-coverage surface. These desorption events occurred as the systems sought to minimise their Gibbs free energy, finding a balance between the charge on the electrode and in the solution phase.

After the initial equilibration phase, no additional H atoms were desorbed, and the presented data were evaluated for each simulation at a constant number of adsorbed H atoms and constant electrode charge. However, the dissolution of H atoms from the surface had important consequences for the determination of the electrode status. Upon dissolution, the H atoms donated part of their electronic charge to the platinum electrode, causing its negative charge to increase, and combined with water, generating three different concentrations of hydronium ions within the solution. As a result, as per ref. 28, the simulated electrodes span a negative range of potential and the three H coverages correspond to different potential ranges, as described in Fig. 4. In our simulations, the electrode's charge does not only originate from the presence of ions in solution but also from the complex interplay between the charging and discharging of interfacial water[28] and adsorbed hydrogen as the potential changes. Thus, although in all simulations contributing to the phase diagram in Fig. 2, the number of Na and Cl ions remains constant at 10 Cl and 10 Na, the observed potential for each coverage fluctuates around an average value. These fluctuations, due to the finite size of our model, allow each single trajectory to explore a continuum range of potentials, from reductive up to above the PZC. The different average values for each coverage are induced by the different surface dipoles induced by different coverages and the different number of H ions in solution. Notably, these fluctuations enable a direct comparison of configurations with different coverage and similar work functions, as it will be explained in detail below, in the subsection 'Statistical Approach'.

https://doi.org/10.1038/s42004-025-01446-w **Article**

In addition, to analyse how local variations in potential affect the distribution of H adsorption sites (H$_{UPD}$ and H$_{OPD}$), short AIMD simulations of ~5 ps (after equilibration) were performed, in addition to the dataset generated using a concentration of 10Na:10Cl, for each hydrogen coverage: variations in ionic concentrations of 10Na:12Cl and 12Na:10Cl were studied for each coverage to generate potentials slightly lower and higher than that associated with the 10Na:10Cl conditions (see Supplementary Fig. S2 and Supplementary Discussion 1).

The presence of charged water, as well as salt and hydronium ions effectively screens the electrode within a few angstrom from the surface. Indeed, the separation between the two electrode surfaces corresponds to at least ~8 Debye screening lengths, so electrolyte bulk conditions are reached in the middle of the cell for each system. The overall cell length in z-direction is influenced by the electrolyte density and the coverage. In order to find the correct length for each coverage we ran classical molecular dynamics simulations in a NPT ensemble. The extracted value for the volume was mediated over a simulation time of the order of microseconds, on top of which a NVT classical simulation was performed, using Gromax code and the Heinz force field[52] to model the metal water interaction. The results are reported in Table 1 for each structure. The large cell cross-section of 365Å$^2$ was chosen to reduce the effects of 2D periodicity along the electrode surface.

## Interfacial potential drop and electrode potential alignment

To understand how hydrogen coverage impacts the structure and behaviour of the Pt-electrolyte interface, it is essential to establish a reliable method for determining the expectation value for H formation energy and coverage for each configuration, as a function of electrode potential, and establish a potential scale, as measured with respect to the PZC.

To this end, we propose a double potential reference method, inspired by Rosmeissl's work[53], as described below.

We first computed the average potential difference between the bulk of the electrolyte and the Fermi level of the electrode (Galvani potential) for each coverage along the full trajectory. To establish a standardised scale for the electrode electrochemical potentials of each configuration and system, we consider the 0 ML structure as representative of the point of zero charge and refer to the electrochemical potential of its Pt electrode to this value. A recent experimental evaluation for planar Pt(111)/electrolyte at a very low ionic (micromolar) concentration and pH 3 reported this as 0.56 V[4].

Next, we compare each electrolyte-platinum potential differences calculated for non-zero coverage, $\Delta\psi$, with that obtained at zero coverage for the PZC, $\Delta\psi_{PZC}$. Notably, the potential drops associated with configurations with different coverage can be compared as all the systems share the bulk electrolyte level, which is used as a reference. The resulting difference of potential drops, $\Delta U$ allows to evaluate the shift from the potential of zero-charge $U_{PZC}$, and hence the absolute voltage, $U$, for the electrodes of each configuration, relative to the PZC. At pH = 0 this reads as:

$$\Delta U = U - U_{PZC} = \Delta\psi - \Delta\psi_{PZC}.$$

the adsorption energy of each of the the non-zero coverage configurations is evaluated as a function of its associated potential, $U$ referred to the PZC, and of pH, similarly to what was proposed by ref. 53:

$$E_{form}(U) = E_{XML}(U) - E_{0ML} - N(E_{H_2}/2 + \Delta U - 2.303 k_B TpH),$$

where $E_{XML}(U)$ is the energy of the non-zero coverage structure, $E_{0ML}$ is the average energy of the bare reference surface, $N$ is the number of hydrogen atoms in the cell, $E_{H_2}$ is the energy of a hydrogen molecule and $\Delta U$ is the potential difference with respect to the RHE.

The main difference between our potential alignment approach and the one developed by ref. 53 lies in the reference level used to evaluate the electrode potential $U$ (and hence $\Delta U$ in our formation energy formula). This paper evaluates the work function for each configuration $\phi_{e^-}$ (measured as the difference between the Fermi level and the electrostatic potential energy

level in front of the solvent) and uses the work function for the electrode at pH = 0 and $U_{SHE} = 0V$, $\phi_{SHE}$, to determine an absolute alignment with the SHE.

$$eU_{RHE} = \phi_{e^-} - \phi_{SHE} + 2.303 k_B TpH$$

This requires introducing constraints in the configuration of water at the interface with vacuum, to avoid forming spurious interfacial dipoles due to limited simulation times. In contrast, we use the Galvani potential energy (difference between the Fermi level and the reference bulk potential energy level of the electrolyte) for a reference configuration for the bare Pt(111)/water interface representing the PZC at pH = 0, $\Delta\psi_{PZC}$, and align this to the RHE via the experimental value for the PZC[4], Exp$_{ref}$ (by doing $U_{PZC_{Exp}} = \Delta\psi_{PZC} - Exp_{ref}$). We then align all other configurations to Exp$_{ref}$ by calculating the differences in Galvani potentials $\Delta\psi$ relative to the PZC reference, $\Delta\psi_{PZC}$

$$e\Delta U_{RHE} = eU_{RHE} - e\,U_{PZC_{Exp}}(pH = 0)$$
$$= \Delta\psi - \Delta\psi_{PZC} + Exp_{ref} + 2.303 k_B TpH - 2.303 k_B T.(pH = 0)$$

therefore our method allows for the determination of the electrode potential without requiring to open a gap in the water bulk.

Notably, the Galvani potential of this configuration is obtained as average along the full trajectory.

The two methods correspond to each other when considering that the difference between the bulk potential of the electrolyte solution and the vacuum in front of it, is a constant surface dipole contribution, which cancels out when evaluating $\Delta U$.

Interestingly, our method also can be mapped into Jun Chen's approach[54], with the difference being in the alignment constant for the reference system at the PZC. In Cheng's case, this is evaluated from the solvation energy of the proton, while in our case, it is taken from experimental datum, Exp$_{ref}$.

Finally, it is important to note that our systems contain on average different concentrations of hydronium ions in bulk electrolyte solution. This variation can impact the position of the reference potential energy level in each of them, given the (relatively) small dimension of ab initio simulation cells. To account for this effect, we introduced a pH correction,— $2.3 k_B TpH$, to the bulk electrolyte solution level. Here, $k_B$ is the Boltzmann constant, $T$ is the temperature, and pH was evaluated from the concentration of hydronium ions in bulk electrolyte solution. In each system, we calculated the concentration of hydronium ions within the same window used for the calculation of the water bulk potential (red box of Fig. 1b).

More specifically, the purpose of this term is to align the reference levels for the bulk solution across all studied metastable states, so that $\Delta\Psi$ can be calculated for every configuration with respect to the same reference, based on the assumption of a solution pH of 0. This alignment correction is essential as the bulk levels used to align different systems need to represent physically equivalent states.

Our alignment correction is calculated as $-2.3 k_B Tln([H^+])$, where $[H^+]$ is the concentration of H$^+$ ions present in the central region of our cell defined above. This correction accounts for the potential difference between the system with the reference bulk solution at pH 0 and the system with the actual measured pH value and charge density in the simulation cell.

In our case, the reference is the bulk electrolyte, related to the potential of zero charge as explained above, via the well-known experimental value versus the SHE [4].

Therefore our method allows for the determination of the electrode potential without requiring to open a gap in the water bulk.

Finally, for the calculation of the electrolyte-platinum potential drop, the electrolyte bulk energy level was obtained with a double mean calculated on the xy averaged potential, with a second sliding window of 10 Å (see Fig. 1b). To identify the value for the bulk electrolyte reference level, we calculated the arithmetical mean of the obtained function on a 4 Å window

in the middle of the solution (red box of Fig. 1b). To identify a value for the bulk platinum level, we used a different double mean, whose second window was tuned to the platinum interlayer distance, to remove the oscillations in the electrode bulk potential.

To analyse the effect of voltage on electrode charge, for each coverage, we computed the Bader charges of the atoms belonging to the electrode and averaged the sum over a number of snapshots taken from the trajectory at regular 500 fs intervals. Our definition of electrode charge included the Bader charges of all Pt atoms, all the adsorbed hydrogen atoms, the chemisorbed water layer and any additional atom within 3 Å from the Pt surface, if present. The voltage associated to each status of charge for the electrode was determined by averaging all potential drops observed during each constant coverage simulation. Since no H adsorption or desorption occurred during the trajectory sampling, each constant coverage simulation is also at constant charge.

## Statistical evaluation of average energy and coverage vs. potential

To evaluate expectation value for the hydrogen adsorption energy as a function of the potential, we compare the energies of different metastable equilibrium states with constant hydrogen coverage and similar Galvani potential and statistically evaluate their occurrence probability. To this end, the data on formation energy, $E_{form}$ from all simulations carried out at different coverages, were binned along the potential axis. Within each bin, all the points from the simulations at different coverages were included. The expectation value for the formation energy at each potential was obtained by reweighting the formation energies by the grand canonical probability

$$p_i = \frac{1}{Z_G} \exp(-E_{form,i}/k_B T),$$

where $Z_G = \sum_i \exp(-E_{form,i}/k_B T)$ is the grand canonical partition function. The data obtained for each bin were then fitted to plot the continuous line of Fig. 2.

The value for the most probable coverage at each potential (Fig. 3a) was obtained using the same re-weighting methodology employed for the formation energy and represented in Fig. 2 by the colour of the formation energy profile.

In particular, we compared the statistical weight of the various metastable coverages, characterised by the same potential drop. It should be noted that the instantaneous values of the coverage at each voltage may not correspond to equilibrium coverage at that voltage. Therefore, due to the limited range of coverages studied, a statistical error is expected in our evaluation.

More specifically, within each bin, the expectation value for the coverage was determined as weighted average by the grand canonical probability $p_i$. The coverage value per bin was then fitted by a B-spline function. The overall coverage and the contributions to it from $H_{UPD}$ and $H_{OPD}$ versus potential are plotted in Fig. 3a. To calculate the contribution of $H_{UPD}$ and $H_{OPD}$ to overall coverage, the relative amount of adsorbed hydrogen atoms in the fcc and top positions was computed for each simulation, and it was assigned to the bin, based on the weighted average. The size of each bin was determined adaptively, ensuring it could produce smooth curves while reproducing all main features of the trend.

## Adsorption/desorption currents

The hydrogen adsorption/desorption current in the limit of a slow sweep rate was obtained using a slightly modified version of Karlberg's approach[8], with the formula

$$I = \pm \nu Q_{tot} \frac{d\theta}{dV}.$$

here, $\theta$ is the coverage, $\pm \nu$ is the sweep rate, assumed to be 50 mV/s, and $Q_{tot}$ is calculated by summing the elementary charge per atom multiplied by the

density of surface Pt atoms as reported by ref. 8 with the extra charge contribution from the applied potential and the interaction with the double layer (Fig. 6), as determined from the Bader analysis. The $H_{UPD}$ and $H_{OPD}$ contributions were calculated in the same way as the coverage.

## Data availability

All data needed to evaluate the conclusions in the paper are present in the paper and/or the Supplementary Information and Supplementary Data 1–8. Additional data related to this paper may be requested from the authors.

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

## Acknowledgements

We are thankful for the computational support offered by Dr. Alin Marin Elena and the useful discussions with Prof. Anthony Kucernak. This work was supported by the Engineering and Physical Sciences Research Council (grant EP/P033555/1). We also gratefully acknowledge the use of the High-Performance Computers at Imperial College London, provided by Imperial College Research Computing Service https://doi.org/10.14469/hpc/2232, and the computing resources provided by STFC Scientific Computing Department's SCARF cluster. The authors gratefully acknowledge the Gauss Centre for Supercomputing e.V. (www.gauss-centre.eu) for funding this project by providing computing time on the GCS Supercomputer SuperMUC-NG at Leibniz Supercomputing Centre (www.lrz.de). This research also used ARCHER2 UK National Supercomputing Service (https://www.archer2.ac.uk), via our membership of the UK's HEC Materials Chemistry Consortium, which is funded by EPSRC (EP/R029431 and EP/X035859). This work was also funded by the Deutsche Forschungsgemeinschaft (DFG, German

Research Foundation) under Germany's Excellence Strategy-EXC 2033-390677874-RESOLV. Finally, we would like to acknowledge the Thomas Young Centre under grant number TYC-101.

## Author contributions

C.S.C. conceived and supervised the research. F.R. and R.K. performed the calculations. All authors (C.S.C., F.R., R.K. and M.S.) conducted and discussed the analyses. F.R. provided a first draft of the manuscript and C.S.C. finalised it for publication. All authors proofread the manuscript.

## Competing interests

The authors declare no competing interests.
