## [Transparent Peer Review file · Communications Chemistry]

This manuscript has been previously reviewed at another Nature Portfolio journal. This document only contains reviewer comments and rebuttal letters for versions considered at Communications Chemistry.

Revealing the Molecular Interplay of Coverage, Wettability, and Capacitive Response at the Pt(111)-Water Solution Interface under Bias

Corresponding Author: Dr Clotilde Cucinotta

Version 0:

Reviewer comments:

Reviewer #1

(Remarks to the Author)

The only serious disagreement is about the role of fluctuations. Naturally, the voltage fluctuates in a canonical ensemble. In contrast, the potential is kept constant in a grand-canonical ensemble.

The authors claim "that the friction parameter influences only the temperature of the trajectory, not the ensemble sampled." However, the fundamental fluctuation-dissipation theorem postulates a mathematical relation between friction (dissipation) and fluctuations.

Fluctuations are deviations from equilibrium. Thus, a fluctuating value of the potential, different from the equilibrium value, will usually not be in equilibrium with the adsorbed hydrogen. So it cannot be used to identify configurations that are in equilibrium with the fluctuating value.

Version 1:

Reviewer comments:

Reviewer #1

(Remarks to the Author)

I am still not convinced of the validity of this procedure, but the authors now explain well what they have done, and the readers may form their own opinion. As I said before, there is much about this article that I like, so in my opinion it can now be published in its present form.

Reviewer #1 (Remarks to the Author):

The only serious disagreement is about the role of fluctuations. Naturally, the voltage fluctuates in a canonical ensemble. In contrast, the potential is kept constant in a grand-canonical ensemble.

The authors claim "that the friction parameter influences only the temperature of the trajectory, not the ensemble sampled." However, the fundamental fluctuation-dissipation theorem postulates a mathematical relation between friction (dissipation) and fluctuations. Fluctuations are deviations from equilibrium. Thus, a fluctuating value of the potential, different from the equilibrium value, will usually not be in equilibrium with the adsorbed hydrogen. So it cannot be used to identify configurations that are in equilibrium with the fluctuating value.

We thank Reviewer 1 for their additional remarks. The reviewer raises an interesting point regarding the potential discrepancy between the instantaneous values of the voltage and the equilibrium coverage at that voltage. We appreciate this observation and would like to clarify that, while we did not explicitly determine the equilibrium coverage at each voltage, we compared the statistical weight of various metastable coverages, each consistent with the same potential drop (and therefore, voltage), using the Boltzmann factor. This approach has been supported in the literature, as highlighted in our previous responses [e.g., Rossmeisl 2016].

This approach involved performing multiple simulations, each at constant coverage, electrode charge and temperature, exploring the canonical ensemble (with corresponding voltage fluctuations) around each coverage. We recognise that a more exhaustive analysis, with finer-grained data points for coverage, would have significantly increased the computational cost, which would have been beyond the scope of this study. Consequently, we acknowledge that the statistical evaluation at each voltage point is subject to an inherent error. To address this, we have added the following paragraph at line 1105 of our manuscript:

"In particular, [To estimate the most probable coverage at each voltage,] we compared the statistical weight of the various metastable coverages, characterised by the same potential drop, [using the Boltzmann factor]. It should be noted that the instantaneous values of the coverage at each voltage may not correspond to equilibrium coverage at that voltage. Therefore, due to the limited range of coverages studied, a statistical error is expected in our evaluation."

We would also like to emphasise that the capability of Car-Parrinello (CP)-like simulations to sample the canonical ensemble has been established in the literature [WIREs Comput Mol Sci 2014, 4:391–406. doi: 10.1002/wcms.1176]. In particular, CP-like scheme ensures that the dissipative contribution in this dynamics is counterbalanced by an additive white noise that obeys to the fluctuation-dissipation theorem. This ensures an accurate canonical sampling of the Boltzmann distribution despite any dissipation, according to the fluctuation-dissipation theorem. Therefore, we believe that each of our simulation trajectories

effectively samples the canonical ensemble around each metastable coverage. A more detailed analysis of the relationship between the value of the friction coefficient contained in the white noise and voltage fluctuations falls beyond the scope of this paper.

We hope that this clarifies the matter and that the paper is now considered acceptable for publication in *Communication Chemistry*.